# The Use of Sodium Hypochlorite at Point-of-Use to Remove Microcystins from Water Containers

**DOI:** 10.3390/toxins13030207

**Published:** 2021-03-12

**Authors:** Matodzi Michael Mokoena, Lutendo Sylvia Mudau, Matlou Ingrid Mokgobu, Murembiwa Stanley Mukhola

**Affiliations:** Department of Environmental Health, Tshwane University of Technology, Staatsartillerie Road, Pretoria West, P/Bag X680, Pretoria 0001, South Africa; mudauls@tut.ac.za (L.S.M.); mokgobumi@tut.ac.za (M.I.M.); mukholams@tut.ac.za (M.S.M.)

**Keywords:** microcystins, point-of-use, sodium hypochlorite, water container, household drinking water

## Abstract

Most conventional water treatment plants are not sufficiently equipped to treat both intracellular and extracellular Microcystins in drinking water. However, the effectiveness of sodium hypochlorite in removing Microcystin in containers at the point-of-use is not yet known. This study aimed to assess point-of-use water container treatment using bleach or sodium hypochlorite (NaOCl) and to assess the health problems associated with microcystins. Thirty-nine percent (29 of 74) of the total selected households were randomly selected to receive and treat their stored container water with sodium hypochlorite. The level of microcystin in the container water was measured after 30 min of contact with sodium hypochlorite. Microcystin concentrations in both the blooming and decaying seasons were higher (mean 1.10, 95% CI 0.46–1.67 µg/L and mean 1.14, 95% CI 0.65–1.63 µg/L, respectively) than the acceptable limit of 1 µg/L in households that did not treat their water with NaOCl, whilst in those that did, there was a significant reduction in the microcystin concentration (mean 0.07, 95% CI 0.00–0.16 µg/L and mean 0.18, 95% CI 0.00–0.45 µg/L). In conclusion, sodium hypochlorite treatment decreased microcystin s to an acceptable level and reduced the related health problems.

## 1. Introduction

One of the first known uses of chlorine for disinfection was in the form of hypochlorite known as chloride of lime [1]. Snow used it in 1850 after an outbreak of cholera in an attempt to disinfect the Broad Street Pump water supply in London [2]. Chlorine is added to water in one of three forms: elemental chlorine (chlorine gas), sodium hypochlorite solution or calcium hypochlorite powder (high-test hypochlorite) [3]. Chlorine gas reacts rapidly with water to form two compounds: hypochlorous acid (HOCl) and hydrochloric acid (HCl). Hypochlorous acid (HOCl) is a weak acid that further dissociates into hypochlorite ions (OCl^−^) and hydrogen ions. These three species exist in an equilibrium that is both pH and temperature dependent; their sum is referred to as the total available chlorine (hydroinstruments) [4]. In a sodium hypochlorite solution, which normally sits at a pH of 11–13, all available chlorine is in the form of hypochlorite ions (OCl^−^), which, as previously discussed, is far less efficacious than hypochlorous acid.

Sodium hypochlorite (NaOCl) is used to treat drinking water during water processing. The targets of sodium hypochlorite treatment include *E. coli* and cyanotoxins, which are present in water due to fecal contamination and production by cyanobacteria, respectively. Cyanobacteria bloom on the surface of the water during the blooming season and die during the decaying season due to unfavorable conditions. Cyanobacteria genera such as *Microcystis, Anabaena, Oscillatoria, Nostoc* and *Anabaenopsis* produce microcystins (hepatotoxins) that are harmful to both animals and humans [5]. The presence of these toxins has effects on both animals and humans if ingested through untreated water or poorly treated water. There are also health effects that can be caused by direct contact with recreational water that is contaminated with these toxins. The USEPA [6] reported that users of water treatment systems can remove cyanobacterial cells and low levels of toxins. However, water systems may face challenges providing drinking water during a severe bloom event, when there are high levels of cyanobacteria and cyanotoxins in drinking water sources.

The World Health Organization (WHO) has reported that most of the world’s water treatment plants are not sufficiently equipped to treat cyanotoxins, with only the most recently commissioned water treatment plants using activated carbon being able to remove microcystins. The occurrence of toxins in drinking water depends on the level of raw water contamination and the water treatment used. An example of auxiliary treatment is the addition of powdered activated carbon (PAC) to remove bad tastes and odors [7]. To remove intracellular toxins, the water treatment system should have either a coagulation-sedimentation-filtration process, which reportedly removes up to 90% of *Microcystis* cells, or a coagulation-DAF (dissolved air flotation) process, which can reportedly remove up to 80% of *Microcystis* cells. USEPA [8] reported that the best treatment for the presence of microcystins is powdered activated carbon (PAC) and granular activated carbon (GAC). To remove extracellular toxins, water treatment systems should have either powder activated carbon adsorption or granular activated carbon, which can reportedly remove up to 85% and 95%, respectively, of extracellular microcystins. Tsuji et al. [9] reported that up to 99% of extracellular Microcystins had been removed by free chlorine. Nicholson et al. [10] and Tsuji et al. [9] studied the chlorination of microcystins using different toxin concentrations and chlorine doses. They found it was possible to destroy microcystins under several conditions. Chlorine destroys microcystins in water mostly at a pH of 6.0; however, deactivation of microcystins was achieved at a pH of 9.0.

The most popular water treatment system that removes toxins from drinking water is reverse osmosis; however, this treatment process is expensive and needs regular maintenance. Proper education of the users of reverse osmosis is needed so that they fully understand how to implement the process at the point-of-use. However, bleach can be used to treat toxins [11,12]. During a cyanobacteria bloom, water is highly turbid, and most water treatment plants are not capable of removing all toxins. Manage et al. [13] reported that microcystins are chemically stable in water and that conventional water treatment processes have failed to reduce chemicals to the levels recommended by the WHO [13,14].

Using flow cytometry, Daly et al. [15] evaluated the effect of chlorine on the cell integrity of toxic *Microcystis aeruginosa* in water from a reservoir. The authors used a higher concentration of chlorine than that which Nicholson et al. [10] suggested. The difference was that Daly et al. [15] first lysed the cells, whilst Nicholson et al. [10] degraded the toxins directly. Tsuji et al. [9] reported that chlorination and ozonation are effective means for the removal of microcystins. Nicholson et al. [10] reported that a chlorine dose of 3 mg/L is effective enough to remove the presence of microcystins in drinking water if a residual of 0.5 mg/L is sustained for 30 min. This study aimed to assess the impact of sodium hypochlorite (NaOCl) on the removal of microcystins at the point-of-use in household containers.

## 2. Results

### 2.1. Microcystin Levels in Water Sources

Water source types, viz. groundwater, communal tap, tank supply and Rand Water, are used daily by communities for their drinking water. Almost of all these sources, however, had some link to the cyanobacteria-contaminated Hartbeespoort Dam water, in which microcystin-producing cyanobacteria cells were reported to be above 65%. Water supplied to all the study areas was extracted from this dam and treated before it was supplied through communal taps and tankers. It was also assumed that those who had groundwater in their yard had a link to dam water, as water infiltrates through the soil and rocks. The control study group from Rand Water supply was assumed to have good quality water, as this water was not linked to the dam. Figure 1 shows the mean microcystin concentration during blooming and decaying seasons. The water samples from the dam and the tap were above acceptable limits (1 µg/L) (red line) in both seasons, whereas microcystins were not observed in either groundwater or Rand Water. Whilst microcystins were observed in tank water samples in the decaying season, they were below the acceptable limits.

Hartbeespoort Dam water is a cyanobacteria breeding area, and most of these bacteria produce microcystin toxins [16]. The presence of these toxins was observed at a mean concentration of 4.3 µg/L in the blooming season, which was reduced to 3.6 µg/L in the decaying season. During the blooming season, cyanobacteria cells multiply and form scum, and the cells experience stress, resulting in the release of toxins into the surrounding area. The same water quality pattern was observed in tap water samples, where, in the blooming season, a mean microcystin concentration of 2.2 µg/L was observed; however, this was reduced by half in the decaying season. Communal tap water samples in the study area were supplied with dam water. The results clearly show that the water treatment process was not decreasing the toxins to an acceptable water level. Tank water also showed the presence of microcystins after the treatment process, as it was possible to measure microcystins even though they were below the acceptable level. This contamination of water sources might have had an impact on the water containers used for drinking water in the households.

### 2.2. Microcystin Levels in Water Containers

Water containers used for drinking purposes were also assessed for the presence of microcystins. Figure 2 shows the results of the mean microcystin concentration measured in both the blooming and decaying seasons. All water samples were found to have less than 1 µg/L of microcystins in all water containers from different water sources in different seasons; however, water container samples from Rand Water in the decaying season were reported to have 1.5 µg/L of microcystins.

These data include the results for bleach-treated and non-treated water. Data for treated water samples from tap and tank samples would have affected the data for non-treated water samples. More details on the bleach-treated and non-treated water samples are presented in Figure 3 and Figure 4.

### 2.3. Microcystin Levels in Water Containers in the Blooming Season

Figure 3 shows the mean microcystin concentration grouped by point-of-use water treatment using sodium hypochlorite (NaOCl). It was observed that the microcystin concentrations from the tap and tank water samples were above the acceptable limits of 1.2 and 1.7 µg/L, respectively. However, there was a significant (r (27) = 0.04, *p* ≥ 0.83) reduction in the water samples from containers that were treated at the point-of-use.

In the decaying season, the only water sources that had more than the acceptable microcystin concentration were Rand Water, tap water and tank water (1.5, 1.3 and 1.2, respectively). After point-of-use water treatment using NaOCl, the levels of microcystins were below the acceptable limits.

As mentioned earlier, microcystins were observed in water containers in both seasons. This is because, during the blooming season, the treatment system operators can act to remove or inactivate cyanotoxins in a number of ways, as supported by USEPA [5]. However, effective management strategies depend on understanding the growth patterns and the species of cyanobacteria dominating the bloom, the properties of the cyanotoxins (i.e., intracellular or extracellular) and the appropriate treatment process. The cells that passed through the water treatment process were able to regenerate inside the containers, as reported by Fosso-Kankue et al. [17]. This process of cell re-growth in water containers re-contaminates the collected and stored containers [18]. Under poor conditions, the cells in water containers die and release toxins that are bound inside their cells. No changes in water quality, in terms of microcystin contamination, were observed between seasons. The levels of microcystins in drinking water containers were not based on seasons, as both favorable and unfavorable conditions support the production of microcystin toxins in the water.

Sodium hypochlorite (NaOCl) is one of the chlorine types used in many households, including for water treatment. Apart from treating or degrading the toxins in water, it was also proven to kill other bacteria, such as *Escherichia coli* and total coliforms. In areas that have cyanobacteria, the use of chlorine, which is inexpensive and readily available, is advisable at the household level as a water treatment to enhance water quality. Communities around the Hartbeespoort Dam use different water sources that are likely to be contaminated by the toxins (microcystins) produced by cyanobacteria. Microcystins are produced by *Microcystis aeruginosa* during blooming and decaying seasons, and these microcystins can survive for at least 21 days under good conditions, such as warm temperatures, rich nutrients and calm wind.

## 3. Conclusions

Microcystins were found to pose a health hazard in the decaying season, which is the time when there are unfavorable conditions. The results of this study also show that appropriately using NaOCl as a point-of-use water treatment can reduce microcystins to acceptable levels (1 and 0.8 µg/L), which would lead to a reduction in the related human health effects; however, further treatment of drinking water is necessary to decrease total toxins to 0 µg/L and reduce the risks of chronic toxicities from exposure to a low concentration of toxins over a long-term period. Education or campaigns on household water treatment focus on outbreaks related to microorganisms, which can be treated by boiling, filtration systems, UV treatment and many other processes. However, in the area in this study where cyanobacteria blooming occurs, not much has been done to educate community members about the health problems caused by cyanobacteria.

## 4. Materials and Methods

### 4.1. Study Area

The study was conducted in the Hartbeespoort Dam area, located in Madibeng Municipality, North West Province, Republic of South Africa, which has a total population of 456,209, making up 98,273 households. The Hartbeespoort Inhabitants Forum (HIF) and the primary investigator were involved in recruiting community members situated around the Hartbeespoort Dam, who were grouped according to their main water source. Community leaders were visited, and the purpose of the study was explained to them; their consent was requested as part of the proper procedure to involve community members. After consent was given by the leaders, participating households were selected randomly. This was done by choosing every second available household with residents so that every household was given an equal opportunity to be selected for the study. All the households selected randomly were marked using a global point system. The primary investigator and one of the community members that was assigned by community leaders made an appointment with a household member to explain the study to them and request the participation of those concerned at a time suitable to them and appropriate for the researcher. The communities involved in the study included privately own houses, Reconstruction and Development Programme (RDP) houses and informal settlements.

### 4.2. Study Population

Seventy-four (74) households were randomly selected from areas surrounding Hartbeespoort Dam, and 39% (29) of them were chosen to receive sodium hypochlorite. The groups were using water from the following sources: communal tap, tank supply, groundwater and Rand Water. The water used by all participating households was tested for the presence of the contaminant in the first months after recruitment; of the households in the study, 39% treated their drinking water by using sodium hypochlorite (NaOCl) to degrade microcystins by up to 99% [9]. This point-of-use water treatment was followed throughout the study.

### 4.3. Point-of-Use Water Treatment Using Sodium Hypochlorite (NaOCl)

The chosen 39% (29) of respondents were provided with sodium hypochlorite (NaOCl) for a year during the study and trained how to treat their water. They were provided with a manual that explained and illustrated all the necessary steps on how to treat their water for drinking purposes. Only one cap or tablespoon of bleach was poured into a full 25 L or 20 L container, and the lid was closed; after mixing the contents, they were instructed to wait for 30–60 min before drinking the water. Covering or closing the containers at all times was emphasized to protect the water from post-contamination.

### 4.4. Sample Collection

Water samples (n = 109) from containers were collected on an established seasonal basis (blooming and decaying seasons) over one year. Duplicate samples were collected from all participating households’ water containers in sterile 500 mL Whirl-Pak bags. Two drops of Lugol’s solution were immediately added to all water samples to prevent further reaction of the chlorine in the samples, and they were kept in a black plastic bag to prevent exposure to sunlight [19]. Immediately on arrival at the laboratory, the water samples were decanted into duplicate 2 mL Eppendorf tubes and frozen at −80 °C until further analysis of the toxins was necessary.

### 4.5. Microcystin Analysis and Interpretation

All samples were assessed for microcystins, which was performed using the Abraxis microcystins-ADDA ELISA kit (microtiter plate) (manufactured by Eurofins Abraxis, Inc., Warminster, PA, USA), from ToxSolutions, kits and Services (Gauteng, South Africa), following the Abraxis procedure (PN.520011). The procedure has six standard solutions and one control. After the mixing, washing and incubation of the microcystin solution according to the Abraxis procedure, the plate was placed into a micro-reader to read the results. All data were captured in a Microsoft Excel spreadsheet, and statistical analysis was done using stata-v10 and SPSS v21.

## Figures and Tables

**Figure 1 toxins-13-00207-f001:**
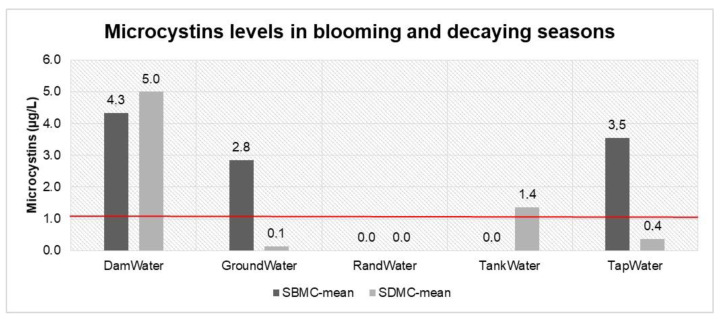
Microcystin water source data grouped by season. SBMC—water source data in the blooming season for microcystins, SDMC—water source data in the decaying season for microcystins. Red line represents the acceptable limits of 1 µg/L according to WHO.

**Figure 2 toxins-13-00207-f002:**
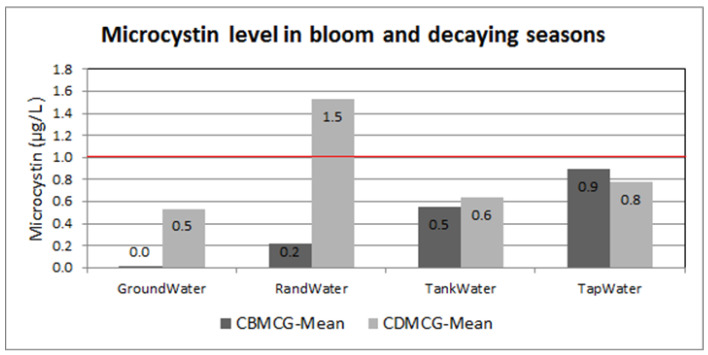
Microcystin water container data, grouped by season. CBMCG—water container data in the blooming season for microcystins in general, CDMCG—water container data in the decaying season for microcystins in general. Red line represents the acceptable limits of 1 µg/L according to WHO.

**Figure 3 toxins-13-00207-f003:**
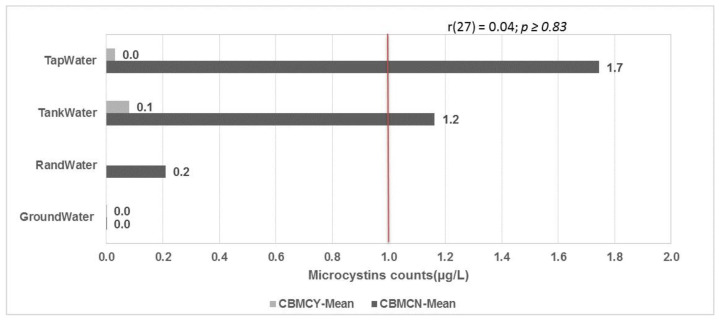
Microcystin water container data during the blooming season, grouped by point-of-use water treatment. CBMCN—water container data for microcystins in the blooming season with no treatment, CDMCY—water container data for microcystins in the blooming season with treatment. Red line represents the acceptable limits of 1 µg/L according to WHO.

**Figure 4 toxins-13-00207-f004:**
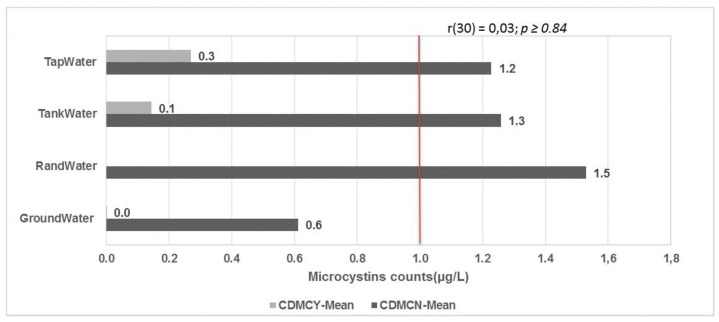
Microcystin water container data during the decaying season, grouped by point-of-use water treatment. CDMCN—water container data for microcystins in the decaying season with no treatment. CDMCY—water container data for microcystins in the decaying season with treatment. Red line represents the acceptable limits of 1 µg/L according to WHO.

## Data Availability

The data presented in this study are available on request from the corresponding author. The data are not publicly available due to that this study is part of Water-related infectious diseases cycle project funded by THRIP funding.

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
