# Peer review of "The Use of Sodium Hypochlorite at Point-of-Use to Remove Microcystins from Water Containers"

_toxins, 2021, doi:10.3390/toxins13030207_

Round 1
Reviewer 1 Report
The manuscript contains publishable data but it deserves to be a little more detailed, especially in the materials and methods part in order to facilitate its reading. Some sentences are missing references in order to justify the ideas and figure 1 is not very clear and above all is not readable without color.
Minor comments:
- Line 6: point-of-point of use ? You mean point-of-use?
- Line 8: The sentence, Thirty-nine per cent (29) of the total households, it is not clear. There is confusion between thirty-nine per cent and the number 29 in parenthesis. It is better to put: Thirty-nine per cent of the total households (29 of 74) par exemple.
- Line 40: the genus Anabaena is actually Dolichospermum.
- Lines 40-42: the sentence "The cyanobacteria genera such as Microcystis, Anabaena, Oscillatoria, Nostoc and Anabaenopsis produce microcystins (hepatotoxins) that are harmful to both animals and human beings" requres adding references.
- Lines 45-47: Thze sentence "The currently used water treatment systems are unable to remove toxins that are produced by cyanobacteria because the toxins are attached in two forms in the cells, viz. intracellular and extracellular toxins" requires adding references.
- Line 67: at pH of 9.0
- Lines 179-181: in the conclusion section it is indicated that "using NaOCl, can reduce the level of microcystins to acceptable levels (1 μg.L-1 and 0.8 μg.L-1), which would lead to reducing the related health effects on humans" THis can be for acute and subchronic toxicities. However, what do you think for chronic exposure to these concentrations.
Major comment:
Lines 155-157: The authors suggested that "As was mentioned earlier, microcystins were observed in water containers in both seasons. This could be because during blooming, cyanobacteria cells can pass through the water treatment process as the water is highly contaminated and the treatment plant was unable to treat all the substances". My question is what the authors did not analyzed intracellular and extracellular microcytins in all samples? This can justify this suggestion.
Reviewer 2 Report
The work is devoted to the experimental removal of toxins from drinking water used from various sources in the same area. This is an important problem especially for dwellings with warm climates and year-round vegetation. The presence of toxins in drinking water can significantly affect the health of the population and the quality of life in general. Therefore, drinking water treatment methods continue to evolve. However, the removal of toxins of blue-green algae, which have nerve-paralytic properties when exposed to the body of animals and humans, is not only an urgent but also a difficult task. The authors experimentally prove that the use of even a small addition of sodium hypochlorite in containers from which the population receives drinking water reduces the level of microcystins by at least ten times, bringing its limits to a level lower than that allowed by the WHO. The work is simply organized, the evidence base is clearly expressed. Statistical processing of analysis data from 106 containers was carried out quite convincingly. The article can be useful not only for those specialists who deal with the problem of removing microcystins from drinking water but also for the administrative authorities responsible for the dissemination of educational information. The article can be published in Water journal with minor changes, most of which are technical and are marked as comments in the attached manuscript file. However, authors should pay attention to some aspects. 1) cyanobacteria are not green algae, otherwise, there would be no problem with toxins in the water, where they cause a bloom. 2) color-coded histograms are very difficult to read, they should be translated into color or in line drawings. 3) The description of the study area should be slightly expanded, only by taking the trouble to look for this area, you can find out which region is in question. 4) in the designation of concentration units, a dot should not be put after micrograms. The rest of the minor remarks are in the text.
